# HCN2 Channels in the Ventral Hippocampal CA1 Regulate Nociceptive Hypersensitivity in Mice

**DOI:** 10.3390/ijms241813823

**Published:** 2023-09-07

**Authors:** Yawen Zheng, Shan Shao, Yu Zhang, Shulu Yuan, Yuanwei Xing, Jiaxin Wang, Xuetao Qi, Kun Cui, Jifu Tong, Fengyu Liu, Shuang Cui, You Wan, Ming Yi

**Affiliations:** 1Neuroscience Research Institute and Department of Neurobiology, School of Basic Medical Sciences, Peking University, Beijing 100191, China; 15369593758@163.com (Y.Z.); shanlihai3@gmail.com (S.S.); 1810305320@pku.edu.cn (S.Y.); xingyw1230@163.com (Y.X.); jiaxinw@bjmu.edu.cn (J.W.); xuetaoqi@pku.edu.cn (X.Q.); 1911110047@pku.edu.cn (K.C.); jifutong@bjmu.edu.cn (J.T.); liufyu@bjmu.edu.cn (F.L.); shuangcs@126.com (S.C.); ywan@hsc.pku.edu.cn (Y.W.); 2National Health Commission Key Laboratory of Human Disease Comparative Medicine, Beijing Engineering Research Center for Experimental Animal Models of Human Critical Diseases, Institute of Laboratory Animal Science, Chinese Academy of Medical Science (CAMS) & Peking Union Medical College (PUMC), Beijing 100101, China; zhangyu@cnilas.org; 3Key Laboratory for Neuroscience, Ministry of Education/National Health Commission, Peking University, Beijing 100101, China

**Keywords:** hyperpolarization-activated and cyclic nucleotide-gated channels, the ventral hippocampal CA1, nociceptive hypersensitivity, hippocampal neuronal excitability

## Abstract

Chronic pain is a significant health problem worldwide. Recent evidence has suggested that the ventral hippocampus is dysfunctional in humans and rodents, with decreased neuronal excitability and connectivity with other brain regions, parallel pain chronicity, and persistent nociceptive hypersensitivity. But the molecular mechanisms underlying hippocampal modulation of pain remain poorly elucidated. In this study, we used ex vivo whole-cell patch-clamp recording, immunofluorescence staining, and behavioral tests to examine whether hyperpolarization-activated cyclic nucleotide-gated channels 2 (HCN2) in the ventral hippocampal CA1 (vCA1) were involved in regulating nociceptive perception and CFA-induced inflammatory pain in mice. Reduced sag potential and firing rate of action potentials were observed in vCA1 pyramidal neurons from CFA-injected mice. Moreover, the expression of HCN2, but not HCN1, in vCA1 decreased in mice injected with CFA. HCN2 knockdown in vCA1 pyramidal neurons induced thermal hypersensitivity, whereas overexpression of HCN2 alleviated thermal hyperalgesia induced by intraplantar injection of CFA in mice. Our findings suggest that HCN2 in the vCA1 plays an active role in pain modulation and could be a promising target for the treatment of chronic pain.

## 1. Introduction

Chronic pain significantly impacts the mental health and work productivity of affected patients, posing a significant public health concern [1,2,3,4]. Even pain with low or moderate intensity may evoke extreme suffering [5]. Nociceptive pain serves as a vital defense mechanism against further injury in response to noxious stress, whereas chronic pain is devastating and induces substantial suffering [6,7]. Current analgesics produce side effects [8,9,10], thus demanding novel molecular targets for pain treatment. Functional and structural abnormalities of the hippocampus underlie the initiation and development of chronic pain [11,12,13,14]. In particular, neurogenesis impairment in the ventral dentate gyrus contributes to CFA-induced inflammatory pain [15], whereas enhancing neuronal excitability of the ventral hippocampal CA1 (vCA1) relieves inflammatory pain [16,17,18].

HCN channels, specifically HCN1 and HCN2 subtypes, serve as regulators of neuronal excitability and are highly expressed in the pyramidal neurons of the hippocampal CA1 [19,20,21,22,23]. HCN channels are activated in response to membrane hyperpolarization [23] and are considered to function as a stabilizing negative feedback mechanism, regulating alterations in membrane potentials [24]. In the hippocampus, HCN channels regulate network excitability by integrating incoming signals, normalizing temporal summation, and facilitating information propagation by attenuating Ca^2+^ signaling [21,25,26]. Indeed, abnormalities of HCN channels in the hippocampus underlie several neurological disorders, such as epilepsy [27].

Recently, there has been increasing attention on the involvement of HCN channels in pain [28,29,30,31]. Both HCN1 and HCN2 channels in the peripheral nervous system participate in nociceptive processing, thereby influencing neuropathic and inflammatory pain [32,33]. In the hippocampus, HCN2 knockdown results in severe degeneration in CA1 pyramidal cell layer [34], whereas knockdown of HCN1 increases the cellular excitability of the hippocampus [35], indicating distinct roles of HCN1 and HCN2 in regulating hippocampal physiology. However, whether HCN channels participate in hippocampal modulation of pain remains unclear. In this study, we examined the intrinsic neuronal properties of vCA1 pyramidal neurons and the HCN channels expression in vCA1 in mice with CFA-induced inflammatory pain, before genetically manipulating the expression level of HCN2 via gene insertion or knockdown in vCA1 pyramidal neurons.

## 2. Results

### 2.1. Decreased Neuronal Excitability and Sag Potential in vCA1 Pyramidal Neurons in Mice with CFA-Induced Inflammatory Pain

To identify the intrinsic neuronal properties of the vCA1 in CFA-injected mice, we conducted targeted whole-cell current-clamp recordings on neurons in the pyramidal layer of vCA1 from CFA modeling mice (Figure 1A). CFA injection reduced the action potentials firing rates in vCA1 pyramidal neurons when subjected to depolarizing current steps (50–400 pA) (Figure 1B; two-way ANOVA; in 200 pA, *p* = 0.045; in 150 pA, *p* < 0.01; two-way ANOVA; interaction: *p* = 0.174; CFA: *p* < 0.01), without affecting other parameters in single action potential (Appendix A). These data indicate that CFA-induced inflammatory pain reduces the excitability of vCA1 pyramidal neurons. In addition, the vCA1 pyramidal neurons in the CFA group showed lower sag responses upon sufficiently strong hyperpolarization than those from control mice (Figure 1C; two-way ANOVA; in 200 pA, *p* < 0.01; in 150 pA, *p* < 0.01; in 150 pA, *p*= 0.031; interaction: *p* = 0.012; CFA: *p* < 0.01). Previous work has shown that opening the HCN channels produces the depolarizing “sag” of membrane potential [24,36], raising the possibility that CFA injection downregulates the activity of HCN channels in vCA1 pyramidal neurons. Furthermore, vCA1 pyramidal neurons in the CFA group showed lower membrane resistance (Figure 1D; *p* = 0.037), also indicating a possible involvement of HCN channels in mediating the decreased excitability of vCA1 neurons.

### 2.2. Decreased Expression of HCN2 in vCA1 of Mice with CFA-Induced Inflammatory Pain

To investigate the potential involvement of HCN channels in the ventral hippocampus in inflammatory pain, we next examined the HCN1/2 expression following CFA injection. One day after the CFA injection, HCN1 and HCN2 protein levels were detected with immunofluorescence and immunoblot, respectively. Expression of HCN1/2 in vCA1 was verified by the number of HCN1/2-positive cells co-labeled with DAPI (nuclear marker) (Figure 2A). The number of HCN2-positive cells and HCN2 protein levels in the vCA1 were significantly reduced in CFA-injected mice (Figure 2B,C; *p* < 0.001). However, no significant alteration was observed in HCN1 levels between CFA and control groups (Figure 2A,B; in immunofluorescence, *p* = 0.261; in Western blot, *p* = 0.549). These data indicate that the expression level of HCN2, but not HCN1, decreases in the ventral hippocampus during the development of CFA-induced inflammation pain in mice.

### 2.3. Up/Down-Regulation of HCN2 in vCA1 Bidirectionally Modulates Nociceptive Hypersensitivity

We next sought to confirm whether HCN2 in the vCA1 affected pain perception in mice. We first knocked down the level of HCN2 in bilateral vCA1 by micro-infusion of shRNA virus and performed pain behavioral tests (Figure 3A and Figure 4A). Virus infection was confirmed through the restricted expression of GFP in vCA1 (Figure 3C), accompanied by reduced HCN2 protein expression (Figure 3D; *p* = 0.032, control vs. down). Four weeks after virus micro-infusion, mice exhibited decreased paw withdrawal latency (PWL), as seen in mice with CFA injection (Figure 4B; *p* < 0.001). Lower PWLs were also observed in mice with HCN2 knockdown in the vCA1 after CFA injection (Figure 4B; *p* = 0.007 on d 3; *p* < 0.001 on d 7; *p* < 0.001 on d 14). As a control, we also downregulated HCN1 in vCA1. However, the knockdown of HCN1 in vCA1 did not affect the PWL of mice (Appendix A), suggesting that only HCN2 in the vCA1 participated in pain modulation. Further, to examine whether overturning the expression level HCN2 in vCA1 could alleviate inflammatory pain, we overexpressed HCN2 with an AAV vector in the vCA1. Elevated HCN2 levels were detected four weeks after virus injection (Figure 3D; *p* < 0.001, control vs. over). As shown in Figure 4C, overexpression of HCN2 in vCA1 significantly increased the PWL (*p* < 0.001) and efficiently attenuated thermal hyperalgesia induced under inflammatory condition (*p* = 0.048 on d 1; *p* = 0.023 on d 3; *p* = 0.020 on d 7; *p* < 0.001 on d 14). Collectively, these findings suggest that HCN2 in the vCA1 affects nociceptive hypersensitivity under both normal and inflammatory conditions.

### 2.4. Down-Regulation of HCN2 Decreased the Excitability of Pyramidal Neurons in vCA1 of Mice

Subsequently, we examined the neuronal excitability in pyramidal neurons after the down-regulation of HCN2 in vCA1 of mice. Four weeks after the shRNA virus infection, we found a significant increase in the count of c-Fos-positive neurons co-labeled with CamkII-GFP-shRNA-HCN2 compared with the control group (c-Fos^+^/GFP-CamkII^+^) (Figure 5A,B; *p* < 0.001), indicating strong inhibition of vCA1 pyramidal neurons with HCN2 knockdown. These pieces of evidence raise the possibility that the down-regulation of the HCN2 expression in vCA1 mediates the thermal hyperalgesia by reducing pyramidal neuronal activity in vCA1.

### 2.5. Over-Expression of HCN2 in vCA1 Has Limited Effect on Anxiety-like Behaviors and Cognitive Impairment in Mice Injected with CFA

Patients experiencing nociceptive pain have an increased vulnerability to the development of anxiety and cognitive impairments [37,38]. To test whether overexpressing HCN2 affected other dimensions of chronic pain, we performed the elevated plus maze test (EPM) and the object–place recognition (OPR) tests to examine the anxiety and cognitive level, respectively. Anxiety-like behaviors and cognitive impairment were observed in mice with chronic inflammatory pain, as evidenced by reduced time spent in the open arms in the EPM test (Figure 6A,B; *p* < 0.001, control vs. CFA) and lower preferences scores in the OPR test (Figure 6C; *p* < 0.001, control vs. CFA). However, overexpressing HCN2 in vCA1 did not affect the time spent (*p* = 0.3393) in the open arms in CFA-injected mice (Figure 6A), nor cognitive and exploratory behaviors of OPR test in mice with CFA injection, indicated by similar preferences (*p* = 0.2092) and total object exploration times (*p* = 0.1901) in the test phase (Figure 6C). In addition, we performed an open field test (OFT) to examine the impact of HCN2 overexpression on locomotion. No statistically significant difference in the total distance traveled in the open field (Appendix A; *p* = 0.0747). These findings suggest that up-regulating HCN2 in vCA1 HCN2 is not capable of fully reversing anxiety-like behaviors and cognitive impairment in mice under inflammatory conditions.

## 3. Discussion

According to the prevailing view, the hippocampus is widely recognized as one of the key brain regions associated with declarative learning and memory. However, recent studies suggest that the hippocampus, its ventral pole in particular, is functionally divergent and plays a role in modulating both neuropathic pain and inflammatory pain [17,39]. One intriguing finding in our experiments is that CFA-induced inflammatory pain results in lower action potential firing rates of vCA1 pyramidal neurons, indicating decreased neuronal excitability. These findings are consistent with previous studies demonstrating suppressed pyramidal neuronal activities in the vCA1 in chronic pain [16,18]. To address the molecular basis of the decreased neuronal excitability of vCA1, we performed a whole-cell patch-clamp which revealed the possible involvement of HCN channels. HCN channels contribute to membrane potential restoration, bringing it closer to the resting membrane potential and generating a hyperpolarizing voltage sag [24,36]. Recent studies point out that pharmacological inhibition or genetic deletion of HCN channels leads to a near-complete elimination of the sag response [40,41], which indicates that the sag potential is the hallmark of HCN channels [42].

In the CFA-induced inflammatory pain model employed in this study, there is a down-regulation of HCN2 expression in the vCA1. Normally, the reduction of HCN channels results in an opposing hyperpolarizing current, which brings anti-excitatory effects to reduce the action potential [23,43]. Considering the crucial role of HCN channels in controlling neuronal firing, abnormal HCN channel expression may contribute to the dysfunctional state of the ventral hippocampus. Our data, therefore, support the hypothesis that reduced HCN2 expression led to decreased neuronal activities in the vCA1 after CFA injection. These findings align with previous studies that have reported severe degeneration of the CA1 pyramidal cell layer following the knockdown of HCN2 expression [34]. Moreover, HCN2-knockin neurons in the thalamic produce bursts of action potentials and reveal a larger voltage–sag ratio [44]. Our c-Fos staining results also indicated a significant reduction in the excitability of pyramidal neurons in the vCA1 after HCN2 knockdown. Considering the limitations of the c-Fos staining, further experiments are necessary to validate the specific effects of HCN2 on the pyramidal cell layer in the vCA1.

It is noteworthy that CFA modeling causes reduced HCN2 but not HCN1 protein levels in the ventral hippocampus. This result possibly links to the more widespread expression of HCN2 in brain regions, including the hippocampus [45]. Similarly, a previous study has confirmed the significant involvement of HCN2 in both inflammatory and neuropathic pain [46], whereas HCN1 channels do not make a significant contribution to heat-induced hyperalgesia [33]. In contrast to our findings, the HCN2 protein expression increases in dorsal root ganglions during persistent inflammation [47], suggesting distinct molecular features of peripheral and central mechanisms of pain modulation [48,49]. In addition, down-regulation of HCN2 in the thalamus and cortex has been reported to provide protection against mechanical allodynia and thermal hyperalgesia in CFA-injected mice [50], further suggesting a distinct pain modulatory role of HCN2 in different brain regions.

We find that up-regulating expression of HCN2 in the vCA1 alleviates nociceptive hypersensitivity, whereas HCN2 knockdown results in hyperalgesia even in naïve mice. Disrupted hippocampal neuronal and network properties have been reported in chronic pain [17,18]. HCN2 channels primarily modulate action potential conduction velocity by modifying the resting membrane potential of neurons [51]. Therefore, we speculate that HCN2 overexpression in the vCA1 leads to an increment of neuronal activity and thereby shows a protective effect on nociceptive hypersensitivity. We also examined anxiety-like and cognitive behavior in mice. The antagonist of HCN channel ZD7288 attenuates neuropathic pain-associated depression [52]. Omrani et al. (2015) [53] also reported that the HCN channel agonist lamotrigine rescued cognitive deficits in mouse models of neurogenetic diseases. In our experiment, however, overexpressing HCN2 in the vCA1 affected the sensory dimension of pain but showed limited influence on anxiety-like behaviors and cognition in models of pain in mice. We need to note that the vCA1 encompasses multiple neuronal subpopulations with different efferent targets [54]. It is not known whether HCN2 expression or its alteration under inflammatory pain is restricted to specific vCA1 neuronal ensembles.

In the present study, we used male but not female mice, since the fluctuation of sex hormones during the menstrual cycle has a strong influence on hippocampal physiology, as well as anxiety and nociception in female mice [55,56,57,58]; the hippocampus also shows gender-related differences in modulating nociceptive hypersensitivity [59]. Considering gender as an important biological variable in nociceptive hypersensitivity could be a potential area for further research.

## 4. Materials and Methods

### 4.1. Animals

Adult male C57BL/6 mice (22–38 g, age 8–12 weeks) were provided by the Department of Laboratory Animal Sciences, Peking University. All experimental animals were kept in plastic cages under standardized conditions, including a controlled temperature of 25 ± 2 °C, a 12 h light/dark cycle, and ad libitum access to standard laboratory chow and tap water. Before behavioral tests, the animals underwent a handling and habituation period of at least 3 consecutive days, during which they were familiarized with the experimenter and the testing environment.

### 4.2. Inflammatory Pain Model

Mice were temporarily anesthetized by 3% isoflurane and received an intraplantar injection of the complete Freund’s adjuvant (CFA, Sigma-Aldrich, St Louis, MI, USA) (40 μL) to the left hind paw to induce persistent inflammatory pain. Mice in the control group received an injection of the same volume of 0.9% saline as a comparison.

### 4.3. Electrophysiological Recordings Ex Vivo

Whole-cell patch-clamp recordings were performed on neurons derived from both the control group and the CFA group. Slices were prepared following the previously described procedure [60]. The mice were anesthetized using isoflurane gas (~2.0%) and sacrificed. Coronal slices of 250 μm thickness, specifically targeting the vCA1 region, were obtained using a vibrating blade microtome (VT1000S; Leica, Germany). The cutting process was performed in an oxygenated ice-cold cutting solution, which consisted of (in mM) 2.5 KCl, 1.25 NaH_2_PO_4_, 0.5 CaCl_2_, 10 MgSO_4_, 26 NaHCO_3_, 10 glucose, and 230 sucrose, pH 7.4, 300–310 mOsm. Next, the coronal sections were incubated for 30 min in a holding chamber containing 32 ℃ artificial cerebrospinal fluid (ACSF) saturated with 95% O_2_ and 5% CO_2_, containing (in mM) 126 NaCl, 2.5 KCl, 1.3 MgCl_2_, 1.2 NaH_2_PO_4_, 2.4 CaCl_2_, 18 NaHCO_3_, and 10 glucose, pH 7.4, 290–300 mOsm.

After incubation, a single slice was transferred to a recording chamber equipped with an Olympus microscope that had infrared digital interference contrast optics. These optics enabled visualization for performing whole-cell patch-clamp recordings. During the entire recording duration, the slices were continuously perfused with ACSF (artificial cerebrospinal fluid) at a rate of 2 mL/min. The temperature was actively maintained at 32 °C in both the holding chamber and during the recording. The patch pipettes were fabricated by pulling borosilicate glass capillary tubes (Sutter 150-86-10, Novato, CA, USA) using a PC-10 pipette puller (Narishige, Tokyo, Japan). The resistance of pipettes varied between 5 and 8 MΩ when filled with a K^+^ Met sulfonate intracellular solution containing (in mM) 140.5 K^+^ Met sulfonate, 7.5 NaCl, 10 4-(2-hydroxyethyl)-1-piperazineethanesulfonic acid (HEPES) hemisodium salt, 2 Mg-ATP, and 0.2 Na-GTP, pH 7.33, 300–310 mOsm.

To evoke a single action potential, a brief current pulse of 500 pA intensity was injected into the cells for a duration of 10 ms. A series of 500 ms current pulses, ranging in intensities from −200 to 400 pA in 50 pA increments, were injected into the cells. The frequencies of the resulting action potentials were recorded. Whole-cell patch-clamp recordings were performed using Multiclamp 700B amplifier (Molecular Devices, Foster City, CA, USA), filtered at 10 kHz, and sampled at 50 kHz. Data were acquired and analyzed using pClamp 10.0 (Axon, Molecular Devices, Foster City, CA, USA).

### 4.4. Immunostaining

Mice were deeply anesthetized using 1% sodium pentobarbital (0.1 g/kg, i.p.) and perfused intracardially with 0.9% saline followed by 4% paraformaldehyde (PFA, in 0.1 M phosphate buffer, pH 7.4). The isolated brain was post-fixed with 4% PFA for 12 h. Subsequently, it was cryoprotected by sequentially immersing it in 20% and 30% sucrose solutions. The fixed brains were then sectioned into 30 µm slices using a cryostat microtome (model 1950, Leica) for immunostaining. Free-floating sections were rinsed with phosphate-buffered saline (PBS) and then blocked using a buffer solution containing 5% bovine serum albumin and 0.3% Triton X-100 for a duration of 1 h, and incubated with primary antibodies at 4 °C for 24 h: rabbit anti-HCN1 primary antibodies (1:200, APC-056, Alomone labs, Jerusalem, Israel); rabbit anti-HCN2 primary antibodies (1:200, APC-030, Alomone labs, Jerusalem, Israel), and rabbit anti-c-Fos (1:500, 2250S, Cell signaling technology, Danvers, USA). Sections were washed with PBS and subsequently incubated with secondary antibodies at room temperature for 60 min: Alexa Fluor 5647-conjugated goat anti-rabbit IgG (1:500, ab150075, Abcam, Cambridge, UK). The brain slices were stained with DAPI (1:1000, 4083S, Cell signaling technology, Danvers, MA, USA). Fluorescence images were captured using a laser scanning confocal microscope (model FV1000, Olympus Co., Ltd., Tokyo, Japan). The numbers of cells were counted by Image-pro Plus 6.0 software (Media Cybernetics, Rockville, MD, USA).

### 4.5. Western Blot

Ventral hippocampal tissues were isolated and homogenized with RIPA lysis buffer (R0010, Solarbio, Beijing, China). Proteins were separated using a 15% SDS-PAGE gel and subsequently transferred onto polyvinylidene fluoride (PVDF) membranes (ISEQ00010, Merck Millipore, Burlington, VT, USA). Membranes were then blocked with 5% non-fat milk for 1 h at room temperature, incubated at 4 °C overnight with primary antibody against HCN1 (1:200, APC-056, Alomone labs Jerusalem, Israel), HCN2 (1:200, APC-030, Alomone labs, Jerusalem, Israel) and β-actin (1:2000, TA-09, ZSGB-Bio, Beijing, China), respectively. After washing three times with tris-buffered saline containing 0.1% Tween 20, the membranes were incubated with appropriate horseradish peroxidase-conjugated secondary antibodies (1:2000, ZB-2301/ ZB-5305, ZSGB-Bio, Beijing, China) at room temperature for 1 h. Protein bands were visualized using a chemiluminescence imaging system (Tanon-5200) and quantified using ImageJ software (version 1.47v; NIH, Bethesda, MD, USA).

### 4.6. HCN1/2 Regulation

To generate AAV vectors, HEK293T cells were transfected with plasmids containing the desired genetic construct along with helper plasmids. After 72 h of transfection, the cells were harvested and lysed through the process of freeze–thawing. The viruses were then isolated and purified by the addition of iodixanol at various concentrations. The genomic titer of each virus was determined using quantitative PCR analysis (Figure 3B, Appendix A). The above process was conducted by the company (Vigene Technology, Jinan, China).

To overexpress HCN1/HCN2, mouse sequences (HCN2: NM_008226.2, HCN1: NM_010408.3) were cloned into the AAV plasmid containing the CamkIIα promoter. Due to technical limitations, the GFP tag could not be added to the AAV plasmid. The DNA cassettes were packaged into recombinant AAV 9 vectors (Titer: 2.55 × 10^13^ vg/mL). The same AAV backbone but only carrying the CamkIIα promoter was used as a control in this study (pAAV-CamkIIα).

To knockdown HCN1/2, shRNA sequences were designed to target mice HCN1/2 transcripts (sense sequence: HCN1: 5′-AGCTGGTTTGTGGTGGACTTCA-3′; HCN2: 5′- GGCATTGTTATTGAGGACAACA-3′) were placed under the CamkIIα promoter. The knockdown efficiency was detected by qPCR to be about 62% and 69% in HCN1/2, respectively. The shRNA was cloned back-to-back with GFP, as a marker of expression. In another plasmid, HCN1/2 shRNA was replaced by a meaningless sequence as a control. The DNA cassettes (containing shRNA and GFP) were packaged into recombinant AAV 9 vectors (Titer: 2.55 × 10^13^ vg/mL).

### 4.7. Stereotaxic Injection

The mice were anesthetized with 1% pentobarbital sodium (35 mg/kg) and subsequently positioned and secured in a stereotaxic instrument (RWD, Shenzhen, China). The scalp was sterilized with iodophors and 75% ethanol, in turn, and incised along the skull midline. Viral vectors (0.3 μL/side) were injected into bilateral vCA1 (AP −3.25 mm; ML ±3.25 mm from bregma; DV −3.25 mm from brain surface). An automatic microinjection system (World Precision Instruments, Sarasota, USA) was used for the injection, with a rate of 0.05 μL/min. After each injection, the needle syringe was held in place for 5 min before withdrawal for solution diffusion. The skin was sutured and then sterilized with iodophors and 75% ethanol. Behavioral tests were performed 3 weeks after the virus injection. Prior to the behavioral tests, the viral infection was confirmed.

### 4.8. Thermal Pain Measurement

Each mouse was habituated to the testing environment before baseline testing. In brief, a focused radiant heat (~15 W of power) produced by Model 336 Analgesic Meter (IITC Inc, Los Angeles, CA, USA) was employed onto the left hind paw. The heat source would be turned off either when the animal withdrew its foot in response to the heat or automatically after 20 s to prevent any potential tissue damage. The time elapsed between the initiation of the thermal stimulation and the withdrawal of the paw was recorded and referred to as the PWL. Experimenters were well-trained to execute consistent behavioral tests and were blinded to the vehicle and treatment groups.

### 4.9. Object–Place Recognition (OPR)

The mice were acclimated to a 40 × 40 × 40 cm box with visual cues for a duration of 20 min per day over a span of 2 days before the actual test. The task consisted of two phases [61]. In the sample phase, mice were placed in the box that contained two identical objects (two cubes, arbitrarily named OA and OB) at two different corners. The mice were given a 10 min period for free exploration before being removed for an additional 20 min. During this time, both the box and objects were thoroughly cleaned using 75% ethanol. Pseudorandomly, one of the two objects (OA) remained unchanged and left in the same place, with the other (OB) moved to a new corner. For the test phase, the mice were reintroduced into the box and given another 10 min period to explore freely. The object exploration was defined when the mouse’s head was <2 cm in distance and pointing towards the object. To avoid object bias, mice with an object preference score greater than 70% or less than 30% during the sample phase were excluded. The preference score was calculated as the exploring time OB/ (OB + OA) in the test phase.

### 4.10. Open Field Test (OFT)

The open field apparatus (50 × 50 × 50 cm) was made of plastic. The mice were given unrestricted movement within the maze, and the behavior of each mouse was recorded for 5 min. We measured the total distance traveled in the field, the total distance traveled, and time spent in the center zone (defined as the square region encompassed by half the diameter) by ezTrack software (version 1.2) [62]. After each test, the inner area of the box was cleaned and sanitized with 75% ethanol and kept for 3 min to dry.

### 4.11. The Elevated Plus Maze Test (EPM)

The EPM consisted of four arms (two open and two closed arms; 30 × 5 cm) crossing at the center perpendicular to each other, forming a plus shape. In the maze, the ‘closed’ arms were surrounded by walls measuring 15 cm in height, providing an enclosed space. On the other hand, the remaining two arms were open without any surrounding walls. To prevent animals from falling off, the open arms were equipped with a 5 mm high border. The platform was positioned at a height of 40 cm above the ground. It was placed in the center of an empty, square tank to ensure the safety of the mouse, preventing any potential falls or attempts to escape during the test. The apparatus was situated in dim light. Every mouse was placed in the central area of the maze, facing toward one of the open arms. Video recordings of the EPM trials were captured using an overhead camera for a duration of 5 min. These recordings were later utilized for behavioral scoring and analysis. After each mouse, the floor was cleaned with ethanol (75%) and dried. Measurements were taken for the following parameters: the number of totally open and closed arm entries, which refers to when all four paws entered an arm; the time spent in the open arms; and the total distance traveled within the arms. Movements were tracked and analyzed using the ezTrack [62] and Matlab 2021a software (version 9.10).

### 4.12. Statistical Analysis

Data are presented as means ± standard error of the mean (SEM). GraphPad Prism (version 8.0, GraphPad Software, La Jolla, CA, USA) was used for statistical analyses. Two-tail unpaired Student’s t-test, ANOVA (one-way, two-way, or repeated measures) with Sidak’s multiple comparisons tests, were used, with *p* < 0.05 as statistically significant.

## 5. Conclusions

CFA-induced inflammatory pain results in decreased pyramidal neuronal excitability and decreased expression of HCN2 in the vCA1. Meanwhile, up/down-regulation of HCN2 in the vCA1 bidirectionally affects thermal sensitivity. Our data reveal a key role of HCN2 in the vCA1 in modulating nociceptive and inflammatory pain.

## Figures and Tables

**Figure 1 ijms-24-13823-f001:**
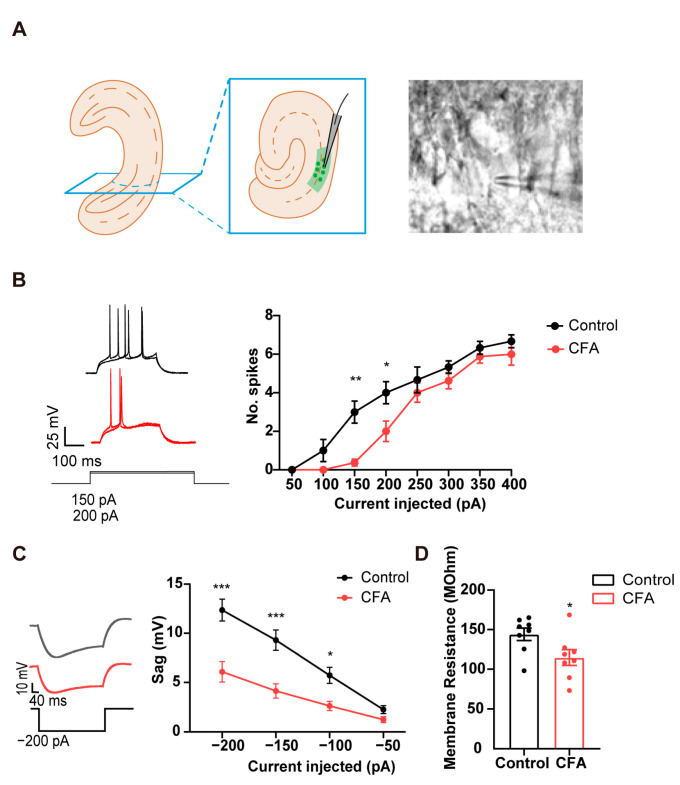
Alterations in electrophysiological properties of vCA1 pyramidal neurons obtained from mice injected with CFA. (**A**) Whole-cell patch-clamp of vCA1 pyramidal neurons. (**B**) Reduced number of elicited action potentials of vCA1 pyramidal neurons in CFA-injected mice. *n* = 8 in each group. * *p* < 0.05, ** *p* < 0.01, two-way ANOVA with Sidak’s multiple comparisons tests. (**C**) The reduced depolarizing sag of vCA1 pyramidal neurons when subjected to hyperpolarizing current pulses in CFA-injected mice. *n* = 8 in each group. * *p* < 0.05, *** *p* < 0.001, two-way ANOVA with Sidak’s multiple comparisons tests. (**D**) Decreased membrane resistance of vCA1 pyramidal neurons in CFA-injected mice. * *p* < 0.05, unpaired two-tailed Student’s t-test.

**Figure 2 ijms-24-13823-f002:**
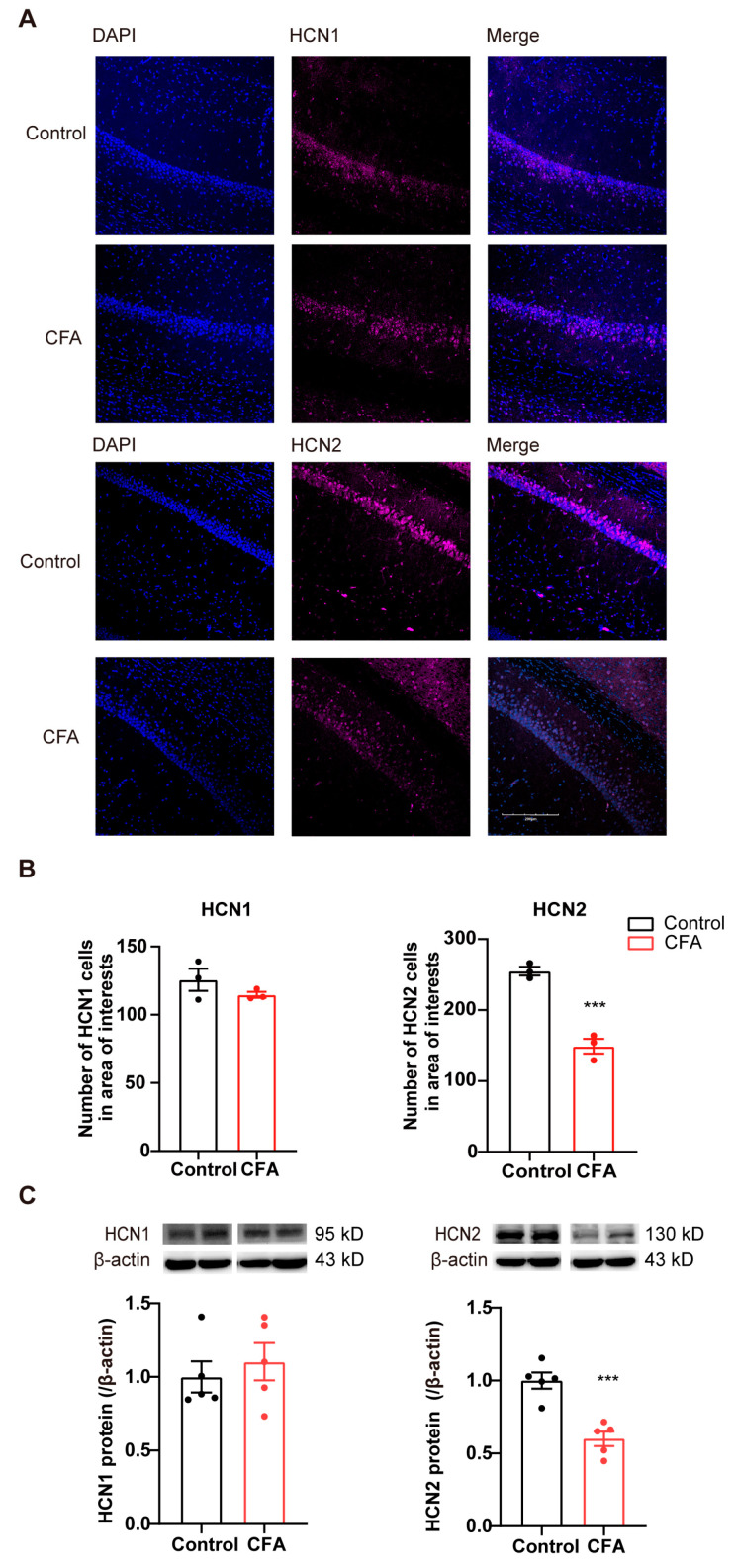
Decreased HCN2 levels in the vCA1 of CFA-injected mice. (**A**) Representative immunofluorescence images showing HCN1 and HCN2 expression in the vCA1. Scale bars, 200 µm. (**B**) Decreased number of HCN2- but not HCN1-positive cells in the vCA1 after CFA injection. *n* = 3 in each group. *** *p* < 0.001, unpaired two-tailed Student’s *t*-test. (**C**) Decreased HCN2, but not HCN1, protein expression levels in the ventral hippocampus after CFA injection. Representative Western blots of HCN1/2 were presented above the corresponding histogram. *n* = 5 in each group. *** *p* < 0.001, unpaired two-tailed Student’s *t*-test.

**Figure 3 ijms-24-13823-f003:**
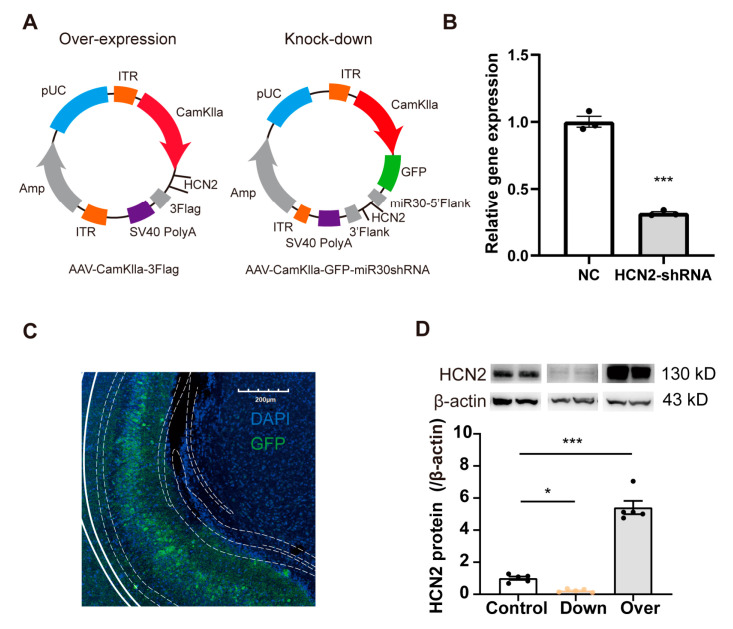
Regulation of HCN2 in vivo and vitro. (**A**) AAV-CamkII virus was injected to overexpress (left) or knockdown (right) HCN2 in the vCA1 of mice. (**B**) Relative levels of HCN1 assessed by real-time RT-PCR in vitro presented a fold change relative to the negative control. *n* = 3 in each group. *** *p* < 0.001 vs. NC, unpaired two-tailed Student’s *t*-test. (**C**) Restricted GFP expression in vCA1. Scale bars, 200 µm. (**D**) Western blot analysis showing altered vCA1 expression of HCN2 after virus injection. Representative Western blots of HCN1/2 were presented above the corresponding histogram. Over: over-expressed HCN2 in vCA1. Down: knocked-down HCN2 in vCA1. *n* = 4 in each group. * *p* < 0.05 vs. control, *** *p* < 0.001 vs. control, unpaired two-tailed Student’s *t*-test.

**Figure 4 ijms-24-13823-f004:**
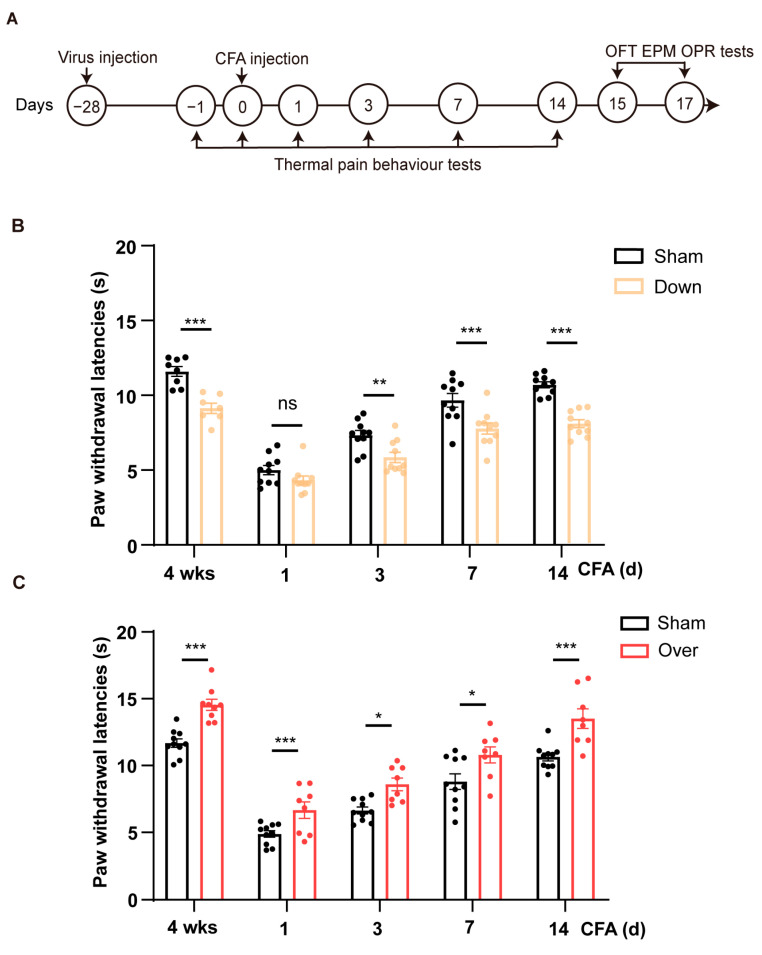
Bidirectional regulation of HCN2 in vCA1 pyramidal neurons modulates nociceptive behaviors in mice. (**A**) The timeline of experiments is depicted in the diagram. Mice were injected with CFA 28 days after virus injection into vCA1. Paw withdrawal latency (PWL) to thermal stimuli was measured before and at specific time points (1, 3, 7, and 14 days) following CFA injection. The elevated plus maze (EPM), (open field test) OFT and object-place recognition (OPR) were tested 15–17 days after CFA injection. (**B**) Knockdown of HCN2 in vCA1 pyramidal neurons reduced thermal pain thresholds. *n* = 8 in each group. ** *p* < 0.01, *** *p* < 0.001, ns: not significant, two-way ANOVA with Sidak’s multiple comparisons tests. (**C**) Overexpressing HCN2 in vCA1 pyramidal neurons attenuated thermal hyperalgesia in mice injected with CFA. *n* = 8 in each group. * *p* < 0.05, *** *p* < 0.001, two-way ANOVA with Sidak’s multiple comparisons tests.

**Figure 5 ijms-24-13823-f005:**
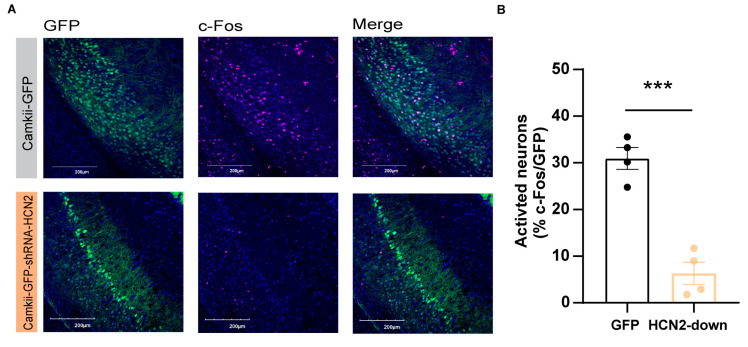
The excitability of vCA1 pyramidal neurons decreased after HCN2 knockdown. (**A**) Confocal images of the vCA1 pyramidal neurons (green) stained with anti-c-Fos(red) and DAPI (blue) in HCN2 knockdown mice. Scale bar: 200 µm. (**B**) The c-Fos expression was decreased after HCN2 knockdown in the vCA1 pyramidal neurons. The activated neurons are calculated as follows: co-labeled c-Fos^+^ (red) and GFP^+^ (green) neurons/GFP^+^ (green) neurons. Three to four mice in either group. *** *p* < 0.001, unpaired two-tailed Student’s *t*-test.

**Figure 6 ijms-24-13823-f006:**
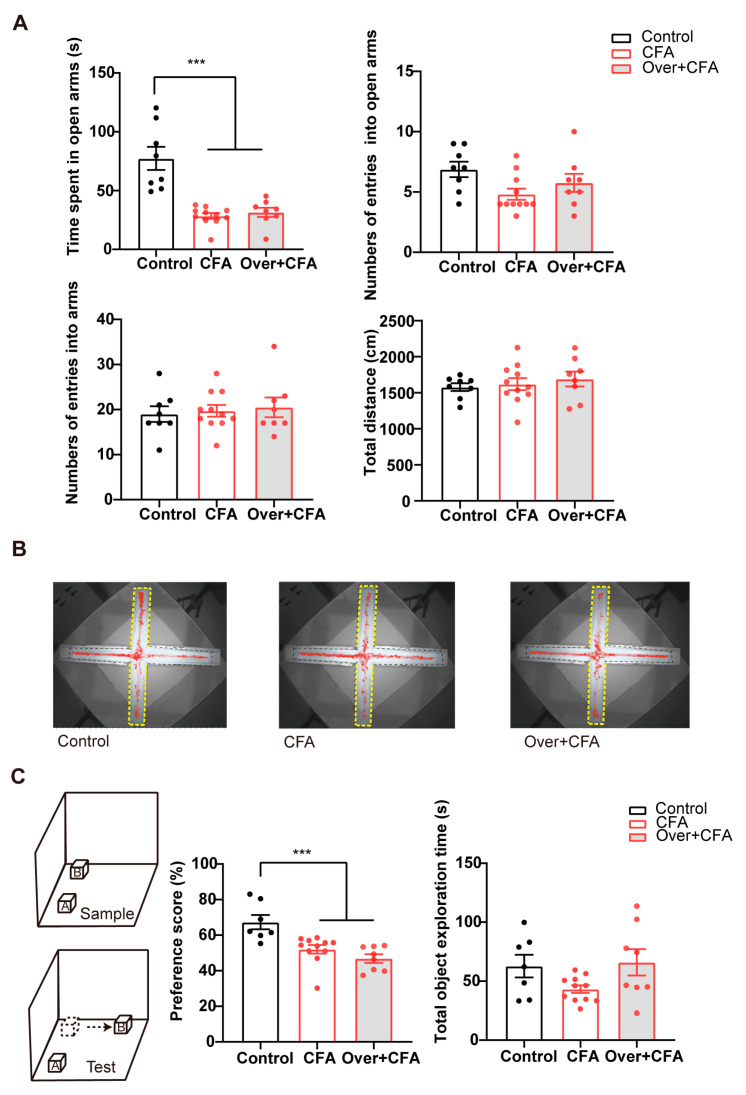
Up-regulation of HCN2 in vCA1 shows limited effects on anxiety-like behaviors and cognitive impairment in CFA-injected mice. (**A**) Up-regulation of HCN2 in vCA1 did not affect time spent and entries into the open arms of the EPM test in CFA-injected mice. No significant changes were observed in the total entries, and distance traveled in the field in all groups. Over + CFA: overexpressed HCN2 in vCA1 of CFA-injected mice. *n* = 8 in each group. *** *p* < 0.001, one-way ANOVA with Sidak’s multiple comparisons tests. (**B**) Representative exploratory tracks (red polylines) in the elevated plus maze. The yellow dashed line indicates open arms in EPM. (**C**) Diagram of the object–place recognition test (left). Up-regulation of HCN2 in vCA1 did not affect the preference score of the OPR test in CFA-injected mice (middle). Total exploration time was similar in all groups (right). *n* = 8 in each group. *** *p* < 0.001, one-way ANOVA with Sidak’s multiple comparisons tests.

## Data Availability

Data pertaining to these studies are available from the corresponding author if and when such a request is deemed appropriate and justified.

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
