# Peer review of "HCN2 Channels in the Ventral Hippocampal CA1 Regulate Nociceptive Hypersensitivity in Mice"

_ijms, 2023, doi:10.3390/ijms241813823_

Round 1

Reviewer 1 Report

Strength:

In this study, authors tried to clarify the role of HCN1 and HCN2 on CFA-induced inflammatory pain in mice. First, it was demonstrated that CFA-induced inflammatory pain reduced the excitability of vCA1 pyramidal neurons by in vivo whole-cell patch clamp technique. Next, the decrement of the expression of HCN2, but not NCN1, was observed in the hippocampus on CFA-induced inflammatory pain in mice. In addition, by the overexpression and downregulation experiment of HCN2 in vCA pyramidal neurons on CFA-induced inflammatory pain in mice, it was observed that HCN2 affected nociceptive hypersensitivity. Finally, it was shown that the overexpression of HNC2 in vCA1 on CFA-induced inflammatory pain in mice almost did not restore the anxiety-like behaviors and cognitive impairment in those mice. These results indicated that HCN in vCA1 may relate to the pain modulation.

Comment:

In Fig.5, authors show the neuronal activity in vCA1 with the c-fos positive cell numbers. However, the c-fos positive cell numbers did not reflect the neuron excitability directly like Fig.1. Please let me know the reason why authors did not examine the decrease of neuron excitability in vCA1 in HCN2-knockdown mice by whole-cell patch clamp technique. In addition, please also explain the reason why authors did not show the neuron excitability in vCA1 in HCN2-overexpression mice.

Please re-check typographical error in whole manuscript.

Reviewer 2 Report

The topic is actual, because chronic pain is a significant health problem worldwide. It has been previously shown that the hippocampus is closely related to the etiopathogenesis of chronic pain. Dysfunctional states of the ventral hippocampus correspond to chronic pain and persistent nociceptive hypersensitivity in both humans and rodents. However, the molecular mechanisms underlying the modulation of pain by the hippocampus are still poorly understood. The authors used modern research methods to identify the involvement of cyclic nucleotide-dependent channels 2 (HCN2) in the ventral CA1 of the hippocampus (vCA1) in the regulation of nociceptive activity. perception and CFA-induced inflammatory pain in mice. The results indicated that HCN2 in vCA1 is actively involved in pain modulation and may be a potential therapeutic target for the treatment of chronic pain. Thus, the article is important both for fundamental science and for practical medicine. The article is written in good language, well-structured and interesting to read.

Author Response

Thank you for carefully reading our manuscript and providing valuable feedback. We are truly grateful for your recognition and affirmation of our research. We have made the revisions to improve our manuscript. Once again, we sincerely appreciate the time and effort you invested in reviewing our manuscript. If you have any further suggestions, we are more than willing to listen and make any necessary improvements.